# Biogeographic Patterns of Leaf Element Stoichiometry of *Stellera chamaejasme* L. in Degraded Grasslands on Inner Mongolia Plateau and Qinghai-Tibetan Plateau

**DOI:** 10.3390/plants11151943

**Published:** 2022-07-26

**Authors:** Lizhu Guo, Li Liu, Huizhen Meng, Li Zhang, Valdson José Silva, Huan Zhao, Kun Wang, Wei He, Ding Huang

**Affiliations:** 1College of Grassland Science and Technology, China Agricultural University, Beijing 100193, China; ellenguo@sina.cn (L.G.); wangkun@cau.edu.cn (K.W.); 2Institute of Grassland, Flowers and Ecology, Beijing Academy of Agriculture and Forestry Sciences, Beijing 100097, China; 3Grassland Research Institute, Chinese Academy of Agricultural Sciences, Hohhot 010010, China; liu_li530@163.com; 4College of Life Sciences, Northwest University, Xi’an 710069, China; 17835417718@163.com (H.M.); xdlizhang@163.com (L.Z.); hewei.scu@gmail.com (W.H.); 5Department of Animal Science, Federal Rural University of Pernambuco, Recife 52171-900, Brazil; valdson.silva@ufrpe.br; 6Academy of Inventory and Planning, National Forestry and Grassland Administration, Beijing 100714, China; zhaohuan333@163.com

**Keywords:** biogeographic patterns, leaf stoichiometry, climatic variables, soil physicochemical properties, *Stellera chamaejasme* L.

## Abstract

Plant leaf stoichiometry reflects its adaptation to the environment. Leaf stoichiometry variations across different environments have been extensively studied in grassland plants, but little is known about intraspecific leaf stoichiometry, especially for widely distributed species, such as *Stellera chamaejasme* L. We present the first study on the leaf stoichiometry of *S. chamaejasme* and evaluate its relationships with environmental variables. *S. chamaejasme* leaf and soil samples from 29 invaded sites in the two plateaus of distinct environments [the Inner Mongolian Plateau (IM) and Qinghai-Tibet Plateau (QT)] in Northern China were collected. Leaf C, N, P, and K and their stoichiometric ratios, and soil physicochemical properties were determined and compared with climate information from each sampling site. The results showed that mean leaf C, N, P, and K concentrations were 498.60, 19.95, 2.15, and 6.57 g kg^−1^; the average C:N, C:P, N:P, N:K and K:P ratios were 25.20, 245.57, 9.81, 3.13, and 3.21, respectively. The N:P:K-ratios in *S. chamaejasme* leaf might imply that its growth is restricted by K- or K+N. Moreover, the soil physicochemical properties in the *S. chamaejasme*-infested areas varied remarkably, and few significant correlations between *S. chamaejasme* leaf ecological stoichiometry and soil physicochemical properties were observed. These indicate the nutrient concentrations and stoichiometry of *S. chamaejasme* tend to be insensitive to variations in the soil nutrient availability, resulting in their broad distributions in China’s grasslands. Besides, different homeostasis strength of the C, N, K, and their ratios in *S. chamaejasme* leaves across all sites were observed, which means *S. chamaejasme* could be more conservative in their use of nutrients improving their adaptation to diverse conditions. Moreover, the leaf C and N contents of *S. chamaejasm* were unaffected by any climate factors. However, the correlation between leaf P content and climate factors was significant only in IM, while the leaf K happened to be significant in QT. Besides, MAP or MAT contribution was stronger in the leaf elements than soil by using mixed effects models, which illustrated once more the relatively weak effect of the soil physicochemical properties on the leaf elements. Finally, partial least squares path modeling suggested that leaf P or K contents were affected by different mechanisms in QT and IM regions, suggesting that *S. chamaejasme* can adapt to changing environments by adjusting its relationships with the climate or soil factors to improve its survival opportunities in degraded grasslands.

## 1. Introduction

Ecological stoichiometry plays an important role in analyzing the composition, structure, and function of a concerned community and ecological system [1,2,3]. Over the last a few decades, one particular focus of ecological stoichiometry has been to document large-scale patterns of and the driving factors for plant carbon: nitrogen: phosphorus (C:N:P) stoichiometry [4,5,6,7,8]. The relationship between leaf stoichiometry, geographic patterns, and climate factors have been studied on both global and regional scales. Geographical variation in foliar ecological stoichiometry is a challenging issue to plant ecologists [5,9,10,11]. Meanwhile, the homeostasis (*H*) of element composition is one of the central concepts of ecological stoichiometry, and its strength is related to the ecological strategy and adaptability of species [2,12]. Stoichiometric homeostasis can help predict the strategies that are used by different plant species to cope with limited resources [2,13]. The nutrient conservatism of high *H*-species could be important mechanism contributing to their success, particularly in natural (unmodified) terrestrial ecosystems, where nutrient supply is often limited and highly variable [14,15]. Indeed, the stoichiometric homeostasis of plants varied with species, growth stages, and element types [16,17,18,19].

*Stellera chamaejasme* L. is a native perennial weed that has distributed abundantly in the alpine meadow on the eastern Tibetan Plateau and typical steppe on eastern Inner Mongolia Plateau of China [20,21]. It competes with forage-grass species for water, nutrition, and space, thereby decreasing the quality of the forage grass and shortening the use of grasslands [22]. The whole plant of *S. chamaejasme* is poisonous and its roots and pollens are most toxic, therefore, livestock may be poisoned by inadvertently inhaling the pollen while grazing [23]. It has become one of the most serious weeds threatening a wide range of grasslands, which were grazed heavily, posing potential hazards to the grassland ecological safety and its impact on animal husbandry sustainability [21]. Previous studies of *S. chamaejasme* focused on its nutrient uptake efficiency and water use efficiency compared to co-existing species [21], its allelochemicals and allelopathic effects on forages [24], and weed control techniques and use [25], but no similar phylogeographical study had ever been conducted on *S. chamaejasme*. Plant nutrient and stoichiometry are key foliar traits with great ecological importance, but previous publications provide limited insight into the biogeographic leaf nutrient and stoichiometry patterns for *S. chamaejasme* [21]. As habitat heterogeneity tends to increase with geographical scale, wide-ranging species can usually use a wide array of resources and tolerate broad environmental conditions or physiological stresses and flourish over a larger area [26,27]. Recent studies have assumed that wide-ranging species always have stronger homeostasis or a weak relationship with nutrient concentrations than narrow-ranging species in response to environmental factors (e.g., soil fertility) [15,26]. The widespread nature of *S. chamaejasme* may be associated with its stoichiometric homeostasis.

Several studies on a regional and global scale reported that changes of the leaf N and P stoichiometry are associated with many biotic and abiotic factors, including climate variables, soil properties, species type, and plant functional groups [4,5,6,7,10,28,29,30]. However, sample collection is commonly limited to a few individuals, a few populations, and averaged at the population or species level, disregarding the intraspecific variability [31]. Investigating the geographic variation within species can help uncover the mechanisms of relationships between plant tissue nutrients and environments [32] by excluding the confounding effects of taxonomic and phylogenetic structure such as those that have been found to influence the geographic patterns in leaf nutrients, and their linkages to climate and soil. Since relationships between environment and plant traits along environmental gradients could be presented as evidence of environmental control over species distribution, examining plant-environment (e.g., climate and soil nutrient availability) interactions may provide some insights into the underlying mechanisms of *S. chamaejasme* distribution in degraded grasslands. However, no studies have yet incorporated information on the geographic patterns in leaf stoichiometry of *S. chamaejasme* in relation to environmental factors.

This study aimed to assess the element stoichiometry of *S. chamaejasme* leaves in degraded grasslands across northern China. The distinct regions of the Qinghai-Tibetan Plateau (QT) and Inner Mongolia Plateau (IM) provide a unique opportunity to test whether there are significant differences in leaf stoichiometry under different environmental conditions and to examine how and to what extent soil and climate modify leaf stoichiometry of *S. chamaejasme* across degraded grasslands. In general, most researchers focused on the roles of C, N, and P stoichiometry in the ecological process from individuals to ecosystems, but potassium (K) is an essential macronutrient that has been partly overshadowed by C, N, and P [5,8,33,34]. Our study also focuses on leaf K concentrations of *S. chamaejasme*, which broadens the contents of ecological stoichiometry. We hypothesized that: (1) *S. chamaejasme*, a wide-spread weed, would exhibit small variation in leaf stoichiometry and tolerate broad environmental conditions; in other words, *S. chamaejasme* may have stoichiometric homeostasis and, (2) due to the differences in limiting factors to vegetation in QT and IM, the relationship between *S. chamaejasme* and environmental factors may be related to different factors in the two regions. To test our hypotheses, we first explored the overall biogeographic patterns of C, N, P, and K stoichiometry of *S. chamaejasme* leaves from 29 sampling sites in the two grassland ecosystems in northern China. We then disentangled the effects of soil and climate on the overall plant stoichiometry pattern and compared the difference between the two regions.

## 2. Results

### 2.1. Pattern of Leaf Ecological Stoichiometry and Soil Physicochemical Properties of S. chamaejasme

Leaf C, N, P, K, and C:N, C:P, N:P, N:K, K:P of *S. chamaejasme* varied little across all the study sites (Table 1 and Appendix A). The mean leaf C, N, P, and K across all sites were 498.60 g kg^−1^, 19.95 g kg^−1^, 2.15 g kg^−1^, and 6.57 g kg^−1^, respectively, and the CV% of leaf P was the largest. Moreover, the mean leaf C:N ratio was 25.20, C:P ratio 245.57, N:P ratio 9.81, N:K ratio 3.13, and K:P ratio 3.21. Inconsistent with the pattern of leaf results, the soil physicochemical properties of *S. chamaejasme*-infested areas varied remarkably (Table 2 and Appendix A). The soil C, N, P, and K exhibited large variations, primarily ranging c. 5.87–84.74 g kg^−1^ for C; 0.24–7.43 g kg^−1^ for N, 0.20–0.82 g kg^−1^ for P, and 0.95–30.55 g kg^−1^ for K. The variation in the soil K content across all the study sites was about 32 times (maximum/minimum), which was the most variable element among the four total elements. The soil mean C:N, N:P, C:P, N:K, and K:P ratios were 13.54, 77.72, 6.34, 0.20, and 38.73, respectively. For the available soil nutrients, soil NN variation was considerably larger than that for the AP, AK, and AN content, as evidenced by coefficients of variation (CVs). Similarly, soil WC, pH, and Ec showed a greater variation throughout the sampling areas.

When comparing the leaf element contents and stoichiometry of *S. chamaejasme* in QT and IM, we found that only the leaf K concentrations, N:K and K:P ratio were significantly different between the two regions (Table 1). Moreover, most soil physicochemical properties were higher in QT than those in IM, except soil AN, NN, and Ec. Specifically, soil P, K, AP, WC, and pH were significantly higher in QT than IM, but soil Ec was significantly lower in QT. Similarly, soil C, N, P, and K stoichiometry showed no significant difference between QT and IM (Table 2).

### 2.2. Ecological Stoichiometry Homeostasis of S. chamaejasme in Degraded Grassland

Pearson correlations analysis indicates that there are only weak or no correlations between leaf ecological stoichiometry and soil physicochemical properties (Appendix A). Furthermore, the relationships between leaf elements and stoichiometry of *S. chamaejasme* and soil by using the homeostasis model were analyzed (Table 3). For C, N (vs. soil N and nitrate N), and K (vs soil K and available K) content, and C:N, N:K, K:P ratios of *S. chamaejasme* leaves were categorized as ‘strictly homeostatic’ (*p* > 0.1). The leaf P content and C:P were ‘weakly plastic’, and the leaf N (vs. soil ammonium N) and N:P ratio were classified as ’weakly homeostatic’.

### 2.3. Spatial Variation of Leaf Elements of S. chamaejasme in Relation to Climatic Factors

No significant relationships among the leaf C and N content and two climatic factors (MAT and MAP) were found using data for all the sample sites or regions (Figure 1). For all the study sites, only the leaf K content was correlated with MAT (*p* < 0.001, Figure 1d). For the regions, it should be noted that in IM, the relationship between the leaf P and climatic factors was significant, but K was not; on the contrary, the K content of *S. chamaejasme* leaves was related to climatic factors but P was not in QT. To be specific, the leaf P concentration increased with increasing MAT and MAP in IM. Moreover, with increasing MAT, leaf K had an increasing trend, but increasing MAP showed an opposite trend in QT.

### 2.4. Relative Roles of Soil and Climatic Factors in Leaf Elements of S. chamaejasme

Variation in the leaf C and P heterogeneity across all the sites was mainly explained by the MAP (C: 46.46%; P: 36.49%; Appendix A). MAT explained a relatively larger percentage of variation in the leaf N heterogeneity (72.62%) and leaf K heterogeneity (51.46%). The interaction of soil, MAP, and MAT explained the different percentage of the variation in four elements (C: 22.16%; N: 6.99%; P: 22.03%; K: 11.85%).

What is more, both the leaf P and K contents of *S. chamaejasme* were affected by soil and climatic factors. Thus, a more in-depth analysis using partial least squares path modeling revealed direct and indirect effects of the environmental drivers on leaf P and K content of *S. chamaejasme* in different regions (Figure 2, Appendix A). Firstly, the influence of climatic factors on soil were bigger in IM than that in QT, and the effect of climatic factors on soil was significant on LP in IM. Secondly, we found that soil factors had a significant effect only on LP in QT only. Thirdly, the effect of climate factors on LP was significant in IM, but the direct effect of climate factors on LP or LK in IM and QT were greater than the indirect. These results suggest that LP or LK were affected by different mechanisms in QT and IM regions. Moreover, the goodness of fit (GOF) was 0.3205 and 0.3556 for LP and LK in QT, respectively, and 0.5490 and 0.4431 in IM. The relatively low predictive power of the model of QT suggested that most variation remained unexplained.

## 3. Discussion

### 3.1. Leaf Ecological Stoichiometry and Soil Physicochemical Properties of S. chamaejasme

It is essential to maintain nutrient elements in sufficient amounts and relatively stable ratios for plants to survive and grow [1,2,35,36,37,38]. This study presents, to our knowledge, the first analysis of leaf element concentrations (C, N, P, K) and ratios (C:N, C:P, N:P, N:K, K:P) of *S. chamaejasme* across degraded grasslands in northern China. Our results show that the leaf C (498.60 g kg^−1^), N (19.95 g kg^−1^), and P (2.15 g kg^−1^) of *S. chamaejasme* were higher than the mean value of all species average in the the China Grassland Transect [38], and that there was no obvious difference between two regions of *S. chamaejasme.* N and P are the most important limiting nutrients for primary productivity in terrestrial ecosystems [39], and a high concentration of N and P in *S. camazepams* leaves suggests its high nutrient uptake efficiency in degraded grasslands, which could facilitate its competitive advantage over other species in nutrient-poor environments. Moreover, K is one of the essential macronutrients that plays a critical role in various metabolic processes, but it has been partly overshadowed in ecological stoichiometry by nitrogen and phosphorus [40,41]. It is worth noting that K concentrations of *S. chamaejasme* were greater in QT than that in IM. The reason may be that the content of nutrients in plants are constrained by nutrient supply in the soil, and the content of soil K is significantly higher in QT, therefore generating this difference. Generally, C:N:P can be used as an effective tool to analyze coupled relationships and differences between each element in the plant-soil system [1,2]. The average leaf C:N and C:P ratio of *S. chamaejasme* were 25.20 and 245.57, respectively, which were lower than the national grassland average leaf C:N (26.86) and C:P (439.84) [38]. The results indicated that *S. chamaejasme* have higher P utilization rates and N utilization efficiency. Previous studies found that nutrient ratios in aboveground vascular plants can be used to distinguish (1) N-limited sites, (2) P- or P+N-limited sites, and (3) K- or K+N-limited sites from each [7,29,42]. The N:P < 14.5, N:K > 2.1, and K:P < 3.4 in *S. chamaejasme* leaf might imply that its growth is restricted by K- or K+N-limited. Both the leaf and soil K content were significantly different between two sampling regions and fertilizer experiments should be conducted to test the validity of this idea in the future.

We found that *S. chamaejasme* could survive in a soil environment with considerable variation, which is consistent with the fact that *S. chamaejasme* is a wide-ranging species in the grasslands of China [22]. The soil conditions for *S. chamaejasme* growth varies considerably from site to site. Soil physicochemical properties varied with a difference of more than 10 times between the maximum and the minimum included C (14.43 times), N (30.94 times), K (32.27 times), NN (26.66 times), and WC (10.60 times), Ec (21.86 times). This may provide a competitive advantage for *S. chamaejasme* against other plant species and help explain its rapid expansion in various environments, even in heavily degraded grasslands. Generally, Tibetan alpine grasslands and Inner Mongolian temperate grasslands, which have different limiting factors, are both zonal grassland types in China [43]. Alpine grasslands are mainly limited by low temperatures in the growing season, while temperate grasslands are affected by drought [38]. Accordingly, our analysis indicated that some soil physicochemical properties of *S. chamaejasme* for the regions were significantly different. Soil WC and pH for Qinghai-Tibet were significantly higher, and the Ec was lower than those for Inner Mongolia. However, apart from SP, SK and SAP, soil C and N concentrations, and other soil available nutrients (AN, NN, AK) for the regions were insignificantly different. These findings suggest that climate imposes important controls on some soil nutrients.

### 3.2. Relationships between Leaf Ecological Stoichiometry and Environmental Variables

Plant nutrient concentrations and their correlations with soil nutrients are considered effective tools for exploring plant adaptation and resource utilization strategies in a severe environment [28,44]. In our study, few significant correlations between leaf ecological stoichiometry of *S. chamaejasme* and grassland soil physicochemical properties were observed, implying an insensitive response to the changes in the soil nutrient supply of *S. chamaejasme*. This supports the finding of Geng et al. [26] and provides confirmation that wide-ranging species are usually able to use a wide array of resources and tolerate broad environmental conditions or physiological stresses, and hence flourish over a larger area. The poor synchronization with local edaphic conditions demonstrates a capacity of *S. chamaejasme* to maintain a high level of function at both high and low resource levels, resulting in their broad distributions in China’s grasslands. Further, stoichiometric homeostasis can help the plants to maintain their element composition at a relatively stable level regardless of changes in nutrient availability via various physiological mechanisms [2,12], and the degree of stoichiometric homeostasis can be indicated by the homeostatic coefficient (*H*) [12,45,46]. Stoichiometric homeostasis had been reported in many dominant palatable species [15,18,47] in grasslands. However, this has not been established in poisonous species. Since unpalatable plants represent the majority of the plant species that were detected after grasslands have been degraded globally [48,49,50], revealing the eco-physiology characteristics of poisonous weeds will help us better understand how the communities that are dominated by poisonous weeds form. Generally, species-level stoichiometric homeostasis was positively correlated with the stability of vegetation [15,18,51]. Meanwhile, the species with the highest degree of N homeostasis consistently had the relatively highest growth rates [19] and well-developed storage systems [15,52]. Therefore, resource utilization and storage functions of these species mitigates environmental variations [53], resulting in spatiotemporal stability in abundance [54]. Our results showed that *S. chamaejasme* leaves contain different homeostasis strength of C, N, and K contents and its ratios, which means *S. chamaejasme* could be more conservative in their use of nutrients improving their adaptation to diverse conditions.

Besides, growing plants induce changes in the composition of soil communities and the physicochemical soil environment [55,56]. A previous study in an alpine meadow ecosystem has shown that *S. chamaejasme* produced more aboveground litter with higher tissue N and lower lignin:N than each of the co-occurring species, and significantly increased the surface soil organic matter [19]. Another study found that *S. chamaejasme* had different ammonia oxidizing bacterial (AOB) that were present with low ammonia oxidation rates, which means greater N availability for *S. chamaejasme* growth due to losses of N reduction [57]. *S. chamaejasme* have positive feedback on soil processes, especially soil C and N, which could be the reason that why leaf C and N contents of *S. chamaejasme* were not different among the two sites of distinct environments, and both were unaffected by local soil factors. Besides, the soil biota plays a pivotal role in modulating primary production by controlling decomposition and nutrient availability, as well as affecting root grazing and plant nutrient uptake [58,59]. Generally, roots of invasive plants enhance or reduce their mutualistic associations with different mycorrhizal fungi or N fixing bacteria [60], which potentially feedback to plant invasion by enhancing N uptake of invaders [61] or by lowering the dependence of plant invaders on arbuscular mycorrhizal fungi (AMF) compared to native species [62]. A recent study suggested that *S. chamaejasme* possessed high ratios of plant growth-promoting proteobacterial endophytes such as *Pseudomonas*, *Acinetobacter,* and *Brevundimonas* [63], and its invaded soils had a lower relative abundance of AMF, but greater pathogenic fungi [64], which in consistent with many invasive plants. Past studies have revealed that *S. chamaejasme* can cultivate the soil environment differently, facilitating it spread rapidly in degraded grassland although it is a native species.

Our results indicate that in the macro scale, leaf C and N do not directly correlate with meteorological factors (MAT and MAP), which is in agreement with previous studies that were conducted in the grassland biomes of China [7]. The weak relationships that was observed between leaf C, N, and climatic variables may result from plant growth, development, metabolism, phenological, and life-history traits rather than from the specific geographic environment. On the contrary, there were close relationships between the leaf P and K and climatic factors (Figure 1). The relationship between the leaf P and climate factors was significant only in IM, and the K content of *S. chamaejasme* leaves was significantly related to climate factors only in QT. We noticed that the correlation of leaf P and MAP (R^2^ = 0.5523) was greater than the relationship between P and MAT (R^2^ = 0.4886) in IM, and the relationship between K and MAT (R^2^ = 0.3338) was greater than that with MAP (R^2^ = 0.2920) in QT. These again reflect the different limiting factors of plant growth in different regions [38]. It is a reasonable assumption that precipitation is a more important limiting factor than the temperature for vegetation growth in arid and semi-arid regions such as Inner Mongolian Plateau temperate grasslands. However, the variation in MAP seems very small and the positive relationships among MAP, soil water content, soil P, and AP were weak (Appendix A). We suggested that altitude changes may the reason behind a positive association between MAP and leaf P in IM considering the ranges of geographical distribution and altitude (N 41.34°~44.77°, E 115.30°~118.16°, 1060 m to 1535 m). Our results for the strong relationships between leaf P, soil P and AP, and soil water content and altitude in IM supported our suggestion (Appendix A). In contrast, the temperature is more likely to have a greater effect on the leaf element concentrations than precipitation in QT alpine grasslands with high-altitude and low temperature. We also found that only leaf K was negatively correlated with MAP in QT. One possible explanation is that K leaches more easily from leaves than N and P, hence it is easy to ascertain the increase of MAP in the studies area leading to more leaf K leaching of *S. chamaejasme*. Another possible explanation may be that K plays many fundamental physiological and metabolic roles in terrestrial plants in relation to water-use efficiency [34]. Some recent studies have observed that K concentrations are related to drought resistance [65,66], therefore, the leaf K content tends to decrease with MAP increase.

Moreover, the MAP or MAT contribution was stronger in leaf elements than soil, which illustrated once more the relatively weak effect of soil physicochemical properties on leaf elements. To explore complex relationships between soil and climatic factors on the leaf P and K contents of *S. chamaejasme*, we conducted a PLS-PM analysis. We found that soil exerted a significant effect on leaf P content and climate affected leaf K directly in QT, while the leaf P content appeared to be limited mainly by climate but the leaf K content was not affected significantly in IM. This does not fit with the fact that climate factors which often affect leaf elements through their influence on soil nutrient status [67]. The arid conditions of the IM may have restricted grassland plants growth by insufficient water supply, but the results of our previous study [21] have proven that high water use efficiency plus high nutrient uptake efficiency of *S. chamaejasme* ensures its competitive advantage on degraded grasslands in IM, which makes the relationship between leaf P or K of *S. chamaejasme* and the soil factors weak in the IM region. However, the leaf P content was positively correlated with soil factors (soil P, available P, nitrate N, and pH), which was not entirely consistent with the result that was obtained in IM. The negative influence of climatic variables on leaf K was significant in QT. This may be the result of the negative relationship between MAP and leaf K, because K shows a greater loss from the plant canopy by foliar leaching than other nutrients such as N and P [34,68]. Our model suggests that the underlying mechanisms behind the leaf P or leaf K content in *S. chamaejasme* were different in the two regions that were studied, which means *S. chamaejasme* developed adjustable relationships with environmental factors to adapt to different growth conditions, thus facilitating its spread in degraded grasslands.

In addition, species’ natural habitats will be subjected to more disturbances in the future due to climate change and habitat degradation that is caused by intensive anthropogenic activities [69,70]. Thus, continuing wide-scale sampling and considering the influence of human activities are required to further develop a deeper understanding of the geographic patterns of leaf stoichiometry in *S. chamaejasme*.

## 4. Materials and Methods

### 4.1. Study Area

A total of 29 sites were selected (10 sites in IM, 19 sites in QT, Figure 3), of which the longitude ranged from 99.68 to 118.16° E and latitude from 33.35 to 44.77° N. The altitudes spanned from 1060 to 3500 m (Appendix A). The mean annual temperature (MAT) and mean annual precipitation (MAP) range from 1.29 to 8.19 °C and 143.84 to 587.53 mm, respectively.

### 4.2. Plant and Soil Sampling

Field measurements were conducted in June–July 2019, when it was the vigorous growth stage for *S. chamaejasme*. At least 30 *S. chamaejasme* plants were randomly collected in each sampling site, then were subdivided into three subsamples, and the leaves of the subsamples were mixed into a composite sample. The samples were ground into fine powder for measuring the content of elements (carbon, nitrogen, phosphorus, potassium). The concentrations of the total C and total N of the *S. chamaejasme* leaves were determined sequentially on a FLASH 2000 elemental analyzer (Thermo Fisher Scientific, Waltham, MA, USA). The total leaf P and K content were determined by using an AA-6300 atomic absorption spectrophotometer (Shimadzu, Japan).

A total of three soil samples (0–15 cm in depth) were collected from each sample site, and each sample was thoroughly mixed with three subsamples and air-dried. The soil samples were removed from the roots and passed through a 100-mesh sieve (30 cm in diameter). Then, the soil was analyzed for soil carbon (C), nitrogen (N), phosphorus (P), potassium (K), ammonium nitrogen (AN), nitrate nitrogen (NN), available potassium (AK), available phosphorus (AP), pH, electrical conductivity (Ec), and water content (WC). The soil physicochemical properties were measured as described by Bao [71], soil C and N by the FLASH 2000 elemental analyzer, K by NaOH fusion-flame photometry, P by NaOH fusion-Mo/Sb colorimetry, soil AN and NN by Auto Discrete analyzer, and soil AK were determined by the flame atomic absorption spectrophotometer. To measure soil AP, the air-dried and pre-weighed soil was extracted using 0.5 mol L^−1^ NaHCO_3_ and the P concentration in the extract was determined by the ammonium molybdate method. The soil pH was measured in a 1:2.5 soil:water suspension, and the soil Ec was measured using a conductivity meter. The soil water contents were determined gravimetrically by oven-drying subsamples at 105 °C for 24 h.

### 4.3. Data Analysis

The means, standard deviations (SD), and coefficients of variation (CV) of the leaf element concentrations and their ratios, and soil physicochemical properties were calculated. Differences between QT and IM were evaluated by Independent-Samples *t*-test. Pearson correlations analysis was used to evaluate the relationship between *S. chamaejasme* leaf ecological stoichiometry and the soil physicochemical properties across the 29 sampling sites.

The degree of homeostasis (*H*) was calculated by plotting the log-transformed values of leaf elements and soil from 29 sites, where the *H* is the inverse of the slope [2]:

Log (leaf element concentration or stoichiometry) = α + (1/H) log (soil element concentration or stoichiometry).

To determine the degree of stoichiometric homeostasis, the method that was proposed by Persson et al. [12] was used. Standardized major axis regression analyses were conducted for C, N, P, K, and C:N:P:K ratios for leaves (The R package ‘smatr 3′) [72]. Since the slope was expected to be ≥ 0, one-tailed *t*-tests with α = 0.1 were used. If the regression was nonsignificant (*p* > 0.1), 1/H was set to zero, and the organism was considered ‘strictly homeostatic’. Species with 1/H = 1 were considered not homeostatic. All the datasets with significant regressions and 0 < H < 1 were categorized as: 0 < 1/H < 0.25: ‘homeostatic’; 0.25 < 1/H < 0.5: ‘weakly homeostatic’; 0.5 < 1/H < 0.75: ‘weakly plastic’; 1/H > 0.75 ‘plastic’. For 1/H > 1, 1/H close to 1 indicates weak or no stoichiometric homeostasis, and 1/H much larger than 1 indicates ‘homeostatic’.

In order to determine the influence of climate factors, we obtained raw daily precipitation and temperature data (2010–2019) from the China Meteorological Administration and calculated the annual precipitation and temperature using the Kriging interpolation method in ArcGIS (ESRI (Environmental Systems Research Institute), Redlands, CA, USA). Therefore, climate data for the mean annual temperature (MAT) and mean annual precipitation (MAP) for the sample sites were obtained. Regression analyses were performed to determine the correlation of the leaf element contents and climate factors (MAT, MAP). Scatter plots were used to visualize the relationships among the leaf element contents and climate factors (MAT, MAP), and liner regression equations were developed.

We determined the relative importance of the soil, MAP, and MAT for leaf C, N, P, and K heterogeneity across all the sites, respectively, using mixed effects models. In these models, soil, MAP, MAT, and their interaction were fitted as fixed factors, and region was fitted as a random factor (The R package ‘lme4′, ‘lmerTest’, ‘glmm.hp’, ‘readxl’, ‘ade4). The soil data (soil C, N, P, K, AP, AK, AN, NN, WC, pH, Ec) were processed by principal component analysis (PCA; The R package ‘Vegan’, ‘FactoMineR’, ‘factoextra’), then the number of first axis was used as the soil parameter (Appendix A; Appendix A). Leaf C, N, P, K, and climatic data (MAP and MATA) heterogeneity were log-transformed to linearized the data.

Partial least squares path modeling (PLS-PM) was employed to explore the direct, indirect, and interactive effects between all the environmental variables for leaf element contents (The R package ‘plspm’). The model included the following variables: Leaf elements (P, K), climate factors (MAT, MAP), and soil factors (K, AK, NN, and pH for leaf P in IM, P, AN and NN for leaf K in IM, P, AP, NN and pH for leaf P in QT, C, K, AP, WC, and pH for leaf K in QT), after testing for collinearity of soil factors with the multivariate analog of *Levene’s* test using the “*betadisper*” function in the vegan package. The indirect effects are defined as multiplied path coefficients between the predictor and response variables, including all the possible paths excluding the direct effect. The final model was chosen among all constructed models based on the Goodness of Fit (GOF) statistic according to the model’s overall predictive power.

## 5. Conclusions

To our knowledge this is the first study to comprehensively research the chemistry of multiple nutritional elements (C, N, P, K) and their ratios in *S. chamaejasme* leaves and its surrounding soil physiochemical properties and quantify the potential controls and variability at a large scale. We found that there was no obvious difference between the leaf C, N, and P content in *S. chamaejasme* from the QT and IM, but the leaf K concentration was significantly higher in QT than that in IM. Inconsistent with the variation of leaf element contents and ratios, the soil physiochemical properties of *S. chamaejasme*-infested areas varied remarkably, and most of them were greater in QT. Our results clearly showed that there was no significant correlation between *S. chamaejasme* leaf ecological stoichiometry and soil physicochemical properties, which supported the fact that the nutrient concentrations and stoichiometry of wide-ranging species tend to be insensitive to variation in soil nutrient availability.

Besides, the different homeostasis strength of C, N, K, and their ratios in *S. chamaejasme* leaves across all the sites were observed, which indicated that *S. chamaejasme* could be more conservative in their use of nutrients improving their adaptation to diverse conditions. Both the C and N content of *S. chamaejasme* leaves were unaffected by any climatic factors, but the leaf P and K were affected differently in QT and IM. Besides, the MAP or MAT contribution was stronger in the leaf elements than soil by using mixed effects models, which illustrated once more the relatively weak effect of soil physicochemical properties on leaf elements. Finally, we conducted a partial least squares path modeling analysis to examine the different effects of soil and climatic on leaf P and K of *S. chamaejasme*. These results suggest that *S. chamaejasme* adapts to changing environments by adjusting its relationships with climate or soil factors to improve their chances of survival and spread in degraded grasslands.

## Figures and Tables

**Figure 1 plants-11-01943-f001:**
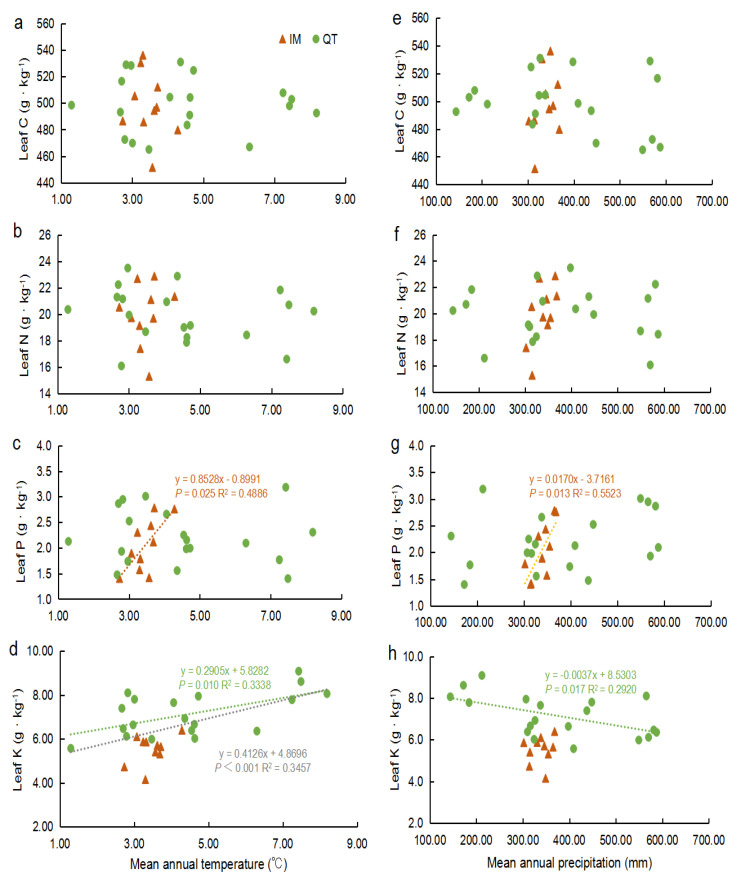
Relationships between the leaf C, N, P, and K content of *S. chamaejasme* with MAT & MAP in the Qinghai−Tibet Plateau (green circles, *n* = 19) and Inner Mongolia Plateau (red triangles, *n* = 10). Linear regression model analyses were utilized. Colored dotted lines represented significant relationships (*p* < 0.05) in different region (red, IM; green, QT; grey, all sampling sites). (**a**) MAT vs. leaf C; (**b**) MAT vs. leaf N; (**c**) MAT vs. leaf P; (**d**) MAT vs. leaf K; (**e**) MAP vs. leaf C; (**f**) MAP vs. leaf N; (**g**) MAP vs. leaf P; (**h**) MAP vs. leaf K.

**Figure 2 plants-11-01943-f002:**
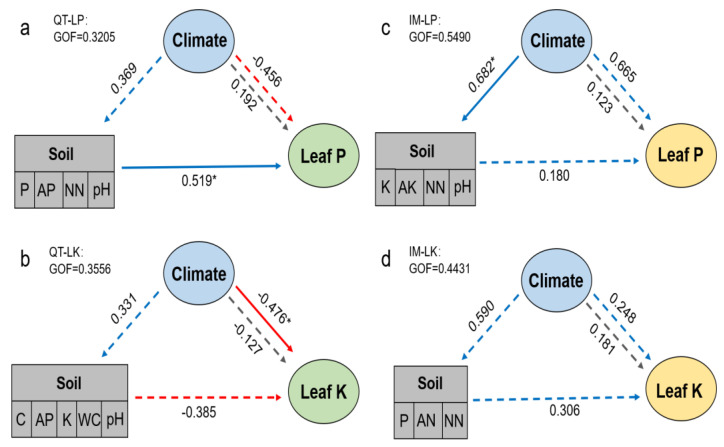
Effects of different soil and climatic variables on the leaf P and K of *S. chamaejasme* in the Qinghai−Tibet Plateau (QT) and Inner Mongolia Plateau (IM) based on partial least squares path modeling. The blue arrows represent positive pathways, the red arrows indicate negative pathways, both are direct effects. The grey arrows show the indirect effects. The standard path coefficients are shown on the arrow. A significant effect is indicated by an * (*p* < 0.05). GOF, goodness of fit of the statistical model. (**a**,**b**) PLS−PM describing the relationships in QT; (**c**,**d**) PLS−PM describing the relationships in IM.

**Figure 3 plants-11-01943-f003:**
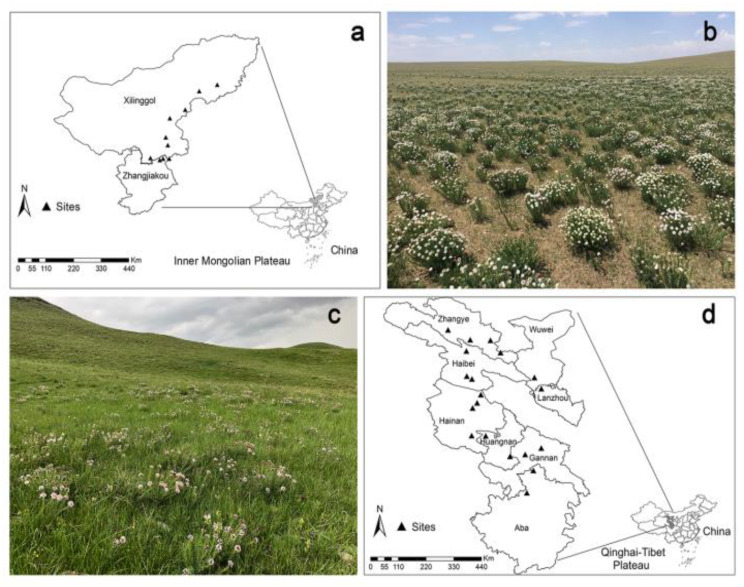
Location of the study and sampling sites. (**a**) Sampling sites on the Inner Mongolia Plateau; (**b**) *S. chamaejasme* coverage in Taipusi Banner in Inner Mongolia Plateau; (**c**) Sampling sites on the Qinghai-Tibetan Plateau; and (**d**) *S. chamaejasme* coverage in Qilian County in Qinghai-Tibetan Plateau.

**Table 1 plants-11-01943-t001:** Regional *S. chamaejasme* leaf ecological stoichiometry. SD is the standard deviation and CV is the coefficient of variation. Differences between QT and IM were tested using independent *t*-test; significant differences at *p* < 0.05 are indicated by different letters.

	All (*n* = 29)		QT (*n* = 19)		IM (*n* = 10)	
	Mean ± SD	CV (%)	Mean ± SD	CV (%)	Mean ± SD	CV (%)
Carbon (g kg^−1^)	498.60 ± 22.07	4.43	498.97 ± 21.14	4.24	497.89 ± 24.91	5.00
Nitrogen (g kg^−1^)	19.95 ± 2.09	10.47	19.94 ± 2.02	10.15	19.97 ± 2.32	11.63
Phosphorus (g kg^−1^)	2.15 ± 0.52	24.33	2.21 ± 0.53	24.15	2.05 ± 0.52	25.20
Potassium (g kg^−1^)	6.57 ± 1.18	17.94	7.13 ± 0.99 a	13.94	5.51 ± 0.66 b	12.02
C:N	25.20 ± 2.27	8.99	25.21 ± 2.18	8.65	25.18 ± 2.543	10.09
C:P	245.57 ± 61.64	25.10	239.45 ± 61.85	25.83	257.20 ± 62.77	24.41
N:P	9.81 ± 2.60	26.54	9.63 ± 2.89	30.05	10.15 ± 2.04	20.05
N:K	3.13 ± 0.63	20.10	2.84 ± 0.45 b	15.70	3.67 ± 0.58 a	15.83
K:P	3.21 ± 0.93	28.89	3.42 ± 1.02 a	29.76	2.81 ± 0.57 b	20.44

**Table 2 plants-11-01943-t002:** Regional *S. chamaejasme* soil physicochemical properties. SD is the standard deviation and CV is the coefficient of variation. Differences between QT and IM were tested using independent *t*-test; significant differences at *p* < 0.05 are indicated by different letters.

	All (*n* = 29)		QT (*n* = 19)		IM (*n* = 10)	
	Mean ± SD	CV (%)	Mean ± SD	CV (%)	Mean ± SD	CV (%)
Carbon (g kg^−1^)	46.11 ± 21.63	46.90	48.65 ± 18.31	37.63	41.29 ± 27.30	66.11
Nitrogen (g kg^−1^)	3.75 ± 1.70	45.24	3.93 ± 1.68	42.70	3.41 ± 1.77	51.85
Phosphorus (g kg^−1^)	0.57 ± 0.17	29.24	0.61 ± 0.13 a	22.04	0.49 ± 0.20 b	40.88
Potassium (g kg^−1^)	20.80 ± 5.86	28.17	22.23 ± 5.00 a	22.49	18.09 ± 6.66 b	36.82
C:N	13.54 ± 6.72	49.62	14.80 ± 7.89	53.30	11.16 ± 2.54	22.91
C:P	77.72 ± 26.00	33.46	79.31 ± 25.53	32.19	74.69 ± 28.01	36.06
N:P	6.34 ± 2.16	34.13	6.25 ± 2.44	39.10	6.52 ± 1.61	25.58
N:P	0.20 ± 0.14	68.81	0.18 ± 0.08	42.44	0.25 ± 0.21	85.03
K:P	38.73 ± 15.12	39.04	38.68 ± 13.27	34.31	38.83 ± 18.95	48.81
Available phosphorus (mg kg^−1^)	5.29 ± 1.96	37.07	5.84 ± 1.90 a	32.57	4.25 ± 1.70 b	40.09
Available potassium (mg kg^−1^)	175.91 ± 96.39	54.79	176.69 ± 106.27	60.14	174.43 ± 79.47	45.56
Ammonium nitrogen (mg kg^−1^)	19.17 ± 7.89	41.14	19.05 ± 7.64	40.12	19.39 ± 8.75	45.11
Nitrate nitrogen (mg kg^−1^)	14.12 ± 14.20	100.59	12.95 ± 4.46	34.42	16.35 ± 24.07	147.25
Water content	0.18 ± 0.08	44.51	0.21 ± 0.08 a	36.46	0.14 ± 0.07 b	53.50
pH	7.90 ± 0.51	6.41	8.05 ± 0.39 a	4.90	7.63 ± 0.60 b	7.89
Electrical conductivity (μs cm^−1^)	247.21 ± 221.21	89.48	193.96 ± 74.42 b	38.37	348.38 ± 351.86 a	101.00

**Table 3 plants-11-01943-t003:** Standardized major axis regression analysis and stoichiometric homeostasis coefficients (*H*) for leaf C, N, P, and K contents and leaf C:N:P:K ratio in *S. chamaejasme* (*n* = 29). All data have been log10-transformed before analysis. If the regression was non-significant (*p* > 0.1), 1/H was set to zero, and the organism was considered to be ‘strictly homeostatic’. Species with 1/H = 1 were considered not homeostatic. All datasets with significant regressions and 0 < H < 1 were categorized as: 0 < 1/H < 0.25: ‘homeostatic’; 0.25 < 1/H < 0.5: ‘weakly homeostatic’; 0.5 < 1/H < 0.75: ‘weakly plastic’; 1/H > 0.75 ‘plastic’. For 1/H > 1, 1/H close to 1 indicates weak or no stoichiometric homeostasis, and 1/H much larger than 1 indicates ‘homeostatic’.

Y	X	1/H (slope)	*p*	r^2^	Category
Leaf C	Soil C	0	0.351	0.0323	strictly homeostatic
Leaf N	Soil N	0	0.829	0.0017	strictly homeostatic
	Soil ammonium N	−0.273	0.089	0.1033	weakly homeostatic
	Soil nitrate N	0	0.291	0.0412	strictly homeostatic
Leaf P	Soil P	0.660	0.042	0.1437	weakly plastic
	Soil available P	0.622	0.002	0.2968	weakly plastic
Leaf K	Soil K	0	0.112	0.0910	strictly homeostatic
	Soil available K	0	0.154	0.0738	strictly homeostatic
Leaf C:N	Soil C:N	0	0.789	0.0028	strictly homeostatic
Leaf C:P	Soil C:P	−0.622	0.018	0.1915	weakly plastic
Leaf N:P	Soil N:P	−0.474	0.085	0.1061	weakly homeostatic
Leaf N:K	Soil N:K	0	0.956	0.0001	strictly homeostatic
Leaf K:P	Soil K:P	0	0.774	0.0031	strictly homeostatic

## Data Availability

Not applicable.

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
