# Peer review of "Biogeographic Patterns of Leaf Element Stoichiometry of Stellera chamaejasme L. in Degraded Grasslands on Inner Mongolia Plateau and Qinghai-Tibetan Plateau"

_plants, 2022, doi:10.3390/plants11151943_

Round 1
Reviewer 1 Report
Manuscript ID: plants-1735226
Type of manuscript: Article
Title: Biogeographic patterns of leaf stoichiometry and adaption strategy of
Stellera chamaejasme L. in degraded grasslands across Northern China
Authors: Lizhu Guo, Li Liu, Huizhen Meng, Li Zhang, Valdson José Silva, Huan
Zhao, Kun Wang, Wei He, Ding Huang * Submitted to section: Plant Ecology,
This is an interesting manuscript relating leaf stoichiometry to biogeographic patterns in Stellera chamaejasme L. across grasslands in China. There are plenty of exciting results in the paper; however, a few issues need to be addressed.
First, the management of the theoretical literature regarding ecological stoichiometry is not deep, and critical issues are missing. For example, given that both C and N were not different among the two sites, there were excluded from ulterior statistical analyses. However, no word as to the causes of why those elements were not different (i.e., their large variability?). Could not these strategies be more critical for the species' widespread distribution? Why were the differences in P or K between sites more important than no differences? Wouldn't the absence of differences reflect higher homeostasis to the more extensive soil elemental ranges?
Because of the weight given to significant statistical differences for the different elements between sites, the interpretation of results remained ambiguous throughout the paper. How is the wide-range distribution of S. chamaejasme explained? Because of the high homeostasis of K or P, the plasticity of C and N, or both? One should acknowledge the limitations of the field assessment of homeostasis (Halvorson & Small, 2016).
Overall, the data in this paper deserves a more comprehensive synthesis and more detailed interpretation based on theoretical grounds (i.e., invasives and invasiveness). In its current version, it is still lacking (see Conclusions).
Lastly, the manuscript should be thoroughly edited in English, as many tenses, incomplete and wordy sentences, and minor typo errors were found.
Figures should be edited to improve clarity on the relationships (Figure 3) and data sources (i.e., differences in the site in Figure 2).
Halvorson, H.M. & G.E. Small.2016. Observational field studies are not appropriate tests of consumer stoichiometric homeostasis. Freshwater Science 35:1103.

Author Response
Dear reviewers,
Thank you for your letter. We were pleased to know that our work was rated as potentially acceptable for publication in Plants, subject to adequate revision. We thank the reviewers for the time and effort that they have put into reviewing the previous version of our manuscript. Their suggestions have enabled us to improve our work. Based on the instructions provided in your letter, we uploaded the file of our revised manuscript (changes in red).
Appended to this letter is our point-by-point response to the comments raised by the reviewers.
We would also like to thank you for allowing us to resubmit a revised copy of the manuscript.
We hope that our revised manuscript is now acceptable for publication in Plants.
Sincerely,
Lizhu Guo

Reviewer 2 Report
Referee comment on “Biogeographic patterns of leaf stoichiometry and adaption strategy of Stellaria chamaejasme L. in degraded grasslands across Northern China”
GENERAL COMMENTS
Guo and colleagues sampled soil and Stellera leaves across the Inner Mongolian and Qinghai-Tibet Plateaus to investigate links among the respective stoichiometries. The authors found that while soil properties varied wildly, leaf nutrient concentrations remained relatively stable across space. No links were found between leaf C, N and the environment, but leaf P and K were related to climate and, to some degree, soil, depending on the region. This is the first study to show substantial nutrient homeostasis for Stellera chamaejasme, which reflects its adaptive strategy and explains among other reasons why this species can occupy such a wide range. The study definitely falls under the category of plant ecology, and there are clear links with the proposed special issue “Ecology of grassland”.
In addition to further comments below, I would like to see the following main points for revision addressed in a next version of this manuscript:
- As someone not familiar with the region, I had to look up that Stellera is actually a native species, but is considered as a (non-invasive?) problematic weed because of its toxicity, domination of the plant community and substantial water consumption. Please clarify better in the manuscript, e.g. in the Introduction, that Stellera is not an invasive exotic species, but nevertheless problematic because of reasons XYZ and that it is colonizing more of the landscape because of reasons ABC. ABC = enhanced dispersal? More disturbances ~ gap colonization? ... Also, it was not clear to me whether degraded grasslands facilitate the establishment/growth of Stellera, or whether Stellera is rather a cause of grassland degradation (it could be both).
- Use of the term “adaptation”: I agree that the ability to maintain tissue stoichiometry across a range of environmental conditions can reflect an adaptation strategy. Use of the term “adaptation” in the title is for example appropriate, in my opinion. However, in some occasions, such as in Line 432 and elsewhere, the reader could be confused and think that the species LOCALLY adapts to the environmental context, i.e. its intraspecific genetics vary across the sites, leading to similar leaf traits for different soil properties. In addition to the species’ inherent capacity to maintain homeostasis irrespective of environment, such adaptation can not be ruled out based on this study, but I think this was not the message you were trying to convey here. Please check the terminology carefully throughout the manuscript to avoid confusion. In addition, how could researchers in a follow-up study assess (quantify) whether stable leaf stoichiometry across environments exists because of local genetic adaptation, or because of inherent capacity of the species to do so?
- The manuscript could benefit from a little bit of a clearer interpretation on nutrient limitation in the regions. Some discussion on P (co-)limitation is already there, but based on the soil and leaf data you have, and for example critical NPK-ratios (sensu Olde Venterink et al., 2003 - Ecology), what can we conclude about nutrient (co-)limitations of this plant across the sites?
- Related to the previous point: given that the title contains the word “stoichiometry”, I would expect more attention to CNPK ratios in the text and especially figures and tables. Why were not ratios shown for plants and soils in Figs. 1, 2 and 3, and Tables S2 and S3?
- Something remarkably missing from the Discussion on leaf vs soil stoichiometry is that not only soils can influence plants, but plants can influence soils. There exists a vast body of literature on so-called plant-soil feedbacks, including on the influence of plants on soils, see e.g. Van Sundert et al., 2021 – European Journal of Forest Research. I suggest at least providing a short paragraph on the influence of plants and soils, why you think we do (not) observe that here and to cite appropriate references.
- The manuscript generally reads very fluently and is very well understandable, but across the manuscript I still found quite some minor errors in grammar and vocabulary. These were not too disturbing but please carefully read through your text again and correct mistakes before submitting a next version of your manuscript. Under “technical corrections” I provide several examples.
SPECIFIC COMMENTS
Line 3 – From the text I understand that Stellera is considered as a problematic species in these regions. However, it is native as far as I understand. If not, you could insert the word “invasive” to the title. Otherwise, specify further in the introduction why the spread/dominance in the plant community of the (native) species is increasing. See also other comments.
Line 42 – Consider adding as keyword “invasive species” or alike, if applicable. The study of invasive species is a whole field in itself and a bit more emphasis on this aspect can attract a broader readership. Stellera is not exotic but is considered nowadays as invasive? See also other comments.
Line 77 – Make clear that this last sentence on Stellera’s homeostasis is a hypothesis.
Line 158 – Remove “however, significant relationships only found in leaf P and K.” : this is obvious from the figure.
Line 158 – “power function regression was utilized”. Please add that this was done, instead of a normal linear regression for example, because it is part of an approach to estimate the degree of homeostasis H.
Line 195 – Remove “however, significant relationships only found in leaf P and K.” : this is obvious from the figure.
Line 241 – “greatest soil physicochemical properties” --> “most extreme”? ~ strong P limitation here?
Line 246 – CV of NN was greater than that of AN, from which you conclude that Stellera was less sensitive to NN variation. I disagree that you can draw this conclusion from these data. Both NN and AN can be taken up by plants, and eventually end up as tissue, including leaf N. No distinction can be made from these data alone to variation in which one the plant stoichiometry would be most sensitive.
Line 275 – “poisonous plants represent the majority of the plant species detected after grasslands have been degraded”. Do you refer here specifically to the kinds of grasslands studied here in China, or is this a general truth globally? Please provide appropriate references for this claim.
Line 278 – Is it useful to calculate H for carbon, given that it is not a nutrient taken up from soil, but an element taken up from the air through photosynthesis? In any case, I do think it is interesting to discuss the (lack of) relationships between plant C and soil C in the context of plant-soil feedbacks, i.e. can it be that soil stoichiometry not only influences that of plants, but also the other way around? See also other comments.
Line 313 – Are you sure that the negative association between MAP and leaf K in QT is because of K leaching from leaves, or are other factors at play here? For example, K plays an important role in stomatal regulation and thus in plant drought tolerance. Perhaps plants growing in drier regions store more K in their leaves. Please check and cite some more literature here to strengthen your claim here, or modify the text accordingly.
Line 336 – This could be the start of an interesting paragraph, but I miss more background on why Stellera is increasing its spread nowadays. Is this because of human-induced disturbance? ~ more opportunities for gap colonization? Changes in dispersal? Other environmental changes? This should be clear from the Introduction through briefly mentioning in a few sentences. Elaborating on Stellara’s “invasiveness” and the link with its evolutionary strategies with respect to stoichiometry can then be discussed further in this Discussion paragraph.
Line 343 – Same question as above. Why is Stellera so common specifically in degraded grasslands?
Line 355 – Were extra precautions needed given the plant’s toxicity?
Line 367 – “100 mesh sieve”: what was the mesh size?
Line 384 – log-transformation given the data distribution: excellent, but then why do we see some figures with linear curves?
Line 405 – Please cite R and the packages correctly. citation(“packagename”)
Line 425 – “wide-ranging species tend to be insensitive to variation in soil nutrient availability”. --> well, perhaps in terms of nutrient concentrations and stoichiometry, as you have shown in this study. But not necessarily in terms of biomass production (Wilfahrt et al., 2021 - Ecosphere). Please specify clearly that you refer to stoichiometry only here.
Line 431 – I would not use the abbreviation “PLS-PM” in the Conclusions, but rather write it out there.
Line 449 – data availability statement not applicable. All summarized data (means +/- SE) are available through the tables, but I strongly suggest making the raw data openly available to facilitate meta-analyses and improve reproducibility.
Figure 2 – Data from both regions seem à priori merged here. Motivate in a reply and in Materials & Methods why this is acceptable here, in contrast to for example figure 3 where you made a distinction.
Figure 3 – In IM, there appears to be a strong positive association between annual precipitation and leaf P. This is briefly mentioned in the discussion and placed in a context of water limitation, but the variation in MAP seems very small – too small to be a good explanation in itself for the influence on leaf P. Please explain in more detail in the discussion what might be going on here. For example, maybe MAP does not tell the whole story on water limitation here? Do sites with lower MAP also have especially low precipitation during the growing season, much lower than at the sites with even a bit higher MAP?
Table 3 shows substantial overlap with Figure 4. I suggest moving the table to the supplement, and adding the significance and indirect pathways to the figure.
Figure 5 – I appreciate very much the inclusion of photos in this figure. It makes clear the differences in vegetation between the two regions during the period of sampling.
Table S2 – In M&M it is mentioned that data were log-transformed because of non-normal distributions. If this was not only the case AMONG sites but also WITHIN sites, then a standard error is not meaningful and I suggest to provide intervals instead, maybe with median. Please clarify in M&M for which analyses, figures and tables data were log-transformed and for which this was not the case.
Table S3 – Please specify n (number of replicates) in the table and figure captions.
TECHNICAL CORRECTIONS
Line 2 – “adaption” should be “adaptation”, I assume.
Line 20 – reflects
Line 28 – sampling site
Line 47 – Remove space before [
Line 70 – “distance” --> “scale”
Line 75 – wide spread
Line 115 – remove “first bullet”
Line 138 – remove double spaces
Line 141 – “we got a very interesting result”. I agree, but this is subjective. Please rephrase.
Line 149 – standard deviation
Line 167 – remove “second bullet”
Line 203 – “Effects of different soil and climatic variables” ...
Line 247 – I suggest to write out NN and AN in full in the Discussion as “soil nitrate” and “soil ammonium”.
Line 276 – revealing
Line 326 – “... soil factors insignificant in the IM region.”
Line 365 – remove one “and”
Line 382 – remove “regions”
Line 422 – “Our results clearly show ...”
Line 435 – “adaptation strategy” .. “in degraded grasslands”
Table 1 – Abbreviations AP, AK, AN, NN, WC and Ec are explained in the Materials and Methods section, but come out of the blue here. Please provide full names in the table caption or in the table itself.
Figure 2 – soil available P, K
REFERENCES
Van Sundert, K., Linder, S., Marshall, J. D., Nordin, A., & Vicca, S. (2021). Increased tree growth following long-term optimised fertiliser application indirectly alters soil properties in a boreal forest. European Journal of Forest Research, 140: 241–254.
Venterink, H. O., Olde Venterink, H., Wassen, M. J., Verkroost, A. W. M., & De Ruiter, P. C. (2003). Species richness-productivity patterns differ between N-, P-, and K-limited wetlands. Ecology, 84: 2191–2199.
Wilfahrt, P. A., Schweiger, A. H., Abrantes, N., Arfin-Khan, M. A. S., Bahn, M., Berauer, B. J., Bierbaumer, M., Djukic, I., van Dusseldorp, M., Eibes, P., Estiarte, M., von Hessberg, A., Holub, P., Ingrisch, J., Schmidt, I. K., Kesic, L., Klem, K., Kroel-Dulay, G., Larsen, K. S., Lohmus, K., Mänd, P., Orban, I., Orlovic, S., Penuelas, J., Reinthaler, D., Radujkovic, D., Schuchardt, M., Schweiger, J. M. I., Stojnic, S., Tietema, A., Urban, O., Vicca, S., & Jentsch, A. (2021). Disentangling climate from soil nutrient effects on plant biomass production using a multispecies phytometer. Ecosphere, 12(8):e03719. 10.1002/ecs2.3719.
Author Response

(The authors gave the same response as above.)

Round 2
Reviewer 2 Report
In my opinion, Guo et al. adequately answered and processed the review comments. I therefore recommend acceptance of this interesting article, after one more minor spell/grammar check by the co-authors.
Author Response
Thank you for your decision and constructive comments on my manuscript.